# S100 Calcium Binding Protein A9 Represses Angiogenic Activity and Aggravates Osteonecrosis of the Femoral Head

**DOI:** 10.3390/ijms20225786

**Published:** 2019-11-18

**Authors:** Re-Wen Wu, Wei-Shiung Lian, Chung-Wen Kuo, Yu-Shan Chen, Jih-Yang Ko, Feng-Sheng Wang

**Affiliations:** 1Department of Orthopedic Surgery, Kaohsiung Chang Gung Memorial Hospital, Kaohsiung 83301, Taiwan; ray4595@gmail.com (R.-W.W.); kojy@cgmh.org.tw (J.-Y.K.); 2Department of Medicine; Graduate Institute of Clinical Medical Sciences, Chang Gung University College of Medicine, Kaohsiung 83301, Taiwan; 3Department of Medical Research, Kaohsiung Chang Gung Memorial Hospital, Kaohsiung 83301, Taiwan; lianws@gmail.com (W.-S.L.); bulakuo@gmail.com (C.-W.K.); ggyy58720240@gmail.com (Y.-S.C.); 4Core Laboratory for Phenomics & Diagnostics, Kaohsiung Chang Gung Memorial Hospital, Kaohsiung 83301, Taiwan

**Keywords:** osteonecrosis of the femoral head, S100 calcium binding protein A9, angiogenesis, vessel, endothelial cells

## Abstract

Ischemic damage aggravation of femoral head collapse is a prominent pathologic feature of osteonecrosis of the femoral head (ONFH). In this regard, S100 calcium binding protein A9 (S100A9) is known to deteriorate joint integrity, however, little is understood about which role S100A9 may play in ONFH. In this study, a proteomics analysis has revealed a decrease in the serum S100A9 level in patients with ONFH upon hyperbaric oxygen therapy. Serum S100A9 levels, along with serum vascular endothelial growth factor (VEGF), soluble vascular cell adhesion molecule-1 (sVCAM-1), interleukin-6 (IL-6), and tartrate-resistant acid phosphatase 5b levels were increased in patients with ONFH, whereas serum osteocalcin levels were decreased as compared to healthy controls. Serum S100A9 levels were increased with the Ficat and Arlet stages of ONFH and correlated with the patients with a history of being on glucocorticoid medication and alcohol consumption. Osteonecrotic tissue showed hypovasculature histopathology together with weak immunostaining for vessel marker CD31 and von Willrbrand factor (vWF) as compared to femoral head fracture specimens. Thrombosed vessels, fibrotic tissue, osteocytes, and inflammatory cells displayed strong S100A9 immunoreactivity in osteonecrotic lesion. *In vitro*, ONFH serum and S100A9 inhibited the tube formation of vessel endothelial cells and vessel outgrowth of rat aortic rings, whereas the antibody blockade of S100A9 improved angiogenic activities. Taken together, increased S100A9 levels are relevant to the development of ONFH. S100A9 appears to provoke avascular damage, ultimately accelerating femoral head deterioration through reducing angiogenesis. This study provides insight into the molecular mechanism underlying the development of ONFH. Here, analysis also highlights that serum S100A9 is a sensitive biochemical indicator of ONFH.

## 1. Introduction

Osteonecrosis of the femoral head (ONFH) is a progressive damage of hip microarchitecture together with catastrophic pain and lower limb disability [1], terribly devastating patients′ activity and even psychology [2], becoming a major cause of hip arthroplasty [3]. Ischemic damage, marrow edema, bone cell death, and femoral head collapse are prominent histopathologic features of the hip disorder. While the underlying cause of ONFH remains uncertain, glucocorticoid overmedication [4], excessive alcohol consumption [5], trauma [6], and genetic variance [7] are known to put hips at risk of the disease.

Decreased blood supply [8] and vascular occlusion [9] occur in the development of human ONFH; however, molecular events underlying the pathological reaction are poorly characterized. In experimental ONFH models, the dephosphorylation of elF2, an endoplasmic reticulum stress regulator, increases angiogenic factor vascular endothelial growth factor (VEGF) production and angiogenesis, slowing femoral head injury during the surgery-mediated ONFH [10]. Epigenetic regulator microRNA-34a is important to inhibit glucocorticoid excess-induced endothelial cell dysfunction and vessel loss in rats with femoral head osteonecrosis [11]. The decreased bone marrow mesenchymal stem cell-derived microRNA-224-3p enhances angiogenic activity in cases of traumatic ONFH [12]. Augmenting vessel remodeling by knee loading attenuates the ligamentum teres transection/ligature around femoral head-induced osteonecrosis [13]. Overactivated toll-like receptor 4 (TLR4) and the nuclear factor-κB (NF-κB) signaling pathways suppress angiogenesis in rats with methylprednisolone and endotoxin lipopolysaccharide-mediated avascular necrosis of the femoral head [14].

The S100 family member S100 calcium binding protein A9 (S100A9) is a 13.2 kDa soluble molecule, which is mostly produced by inflammatory cells and injured tissue in various pathological contexts [15]. TLR4 mediates the S100A9 aggravation of the inflammatory reaction and ischemia/reperfusion-mediated cardiovascular damage [16,17]. The inhibition of TLR4 signaling in vessel endothelial cells attenuates S100A9-induced angiogenesis loss [18] through NF-κB, the signal transducer and activator of transcription 3 (STAT3), and mitogen-activated protein kinase (MAPK) pathways [19]. Increasing evidence has revealed that S100A9 mediates the polarization of bone-marrow macrophages, decreasing muscle vascular regeneration upon ischemic stress [20]. This molecule also modulates vascular thrombosis [21,22] and glomerulonephritis dysregulation of vessel formation [23]. In addition, accumulating evidence has revealed the relevance of S100A9 to arthritic diseases, like osteoarthritis, rheumatoid arthritis, and psoriatic arthritis [24]. Mice deficient in S100A9 show mild osteoarthritis syndrome upon destabilized medial meniscus-meniscus surgery as compared to wild-type mice [25]. The pharmacological inhibitor of S100A9, paquinimod, attenuates the collagenases-mediated synovitis, cartilage loss, and osteophyte formation [26]. The contribution of S100A9 to the development of ONFH remains elusive.

This study aims to investigate whether serum S100A9 is relevant to human ONFH and reveal that role which S100A9 may play in vessel formation in femoral head osteonecrotic tissue.

## 2. Results

### 2.1. Decreased Serum S100A9 Levels in ONFH upon Hyperbaric Oxygen Therapy

First, we investigated the proteomics profiles (Figure 1A) of nucleated cells in peripheral blood in 5 patients with Ficat and Arlet stage II of ONFH before and after 2.5 ATA hyperbaric oxygen therapy (HBO; 1 treatment/day) for 20 treatments [27,28]. Eight proteins homologues to filamin A (FLNA), gelsolin isoform B, disulfide isomerase-associated 3 isoform 1, annexin III chain III, and capping protein (actin filament) muscle Z-line-β were increased upon HBO therapy, whereas 3 proteins homologues to actin beta (ACTB) protein and S100A9 were reduced (Figure 1B), as evident from the tandem mass spectrometry (Figure 1C and Table 1). Of the proteins, S100A9, a 13.2 kDa and pI 5.71 molecule (Figure 1D), is known to regulate joint remodeling [23,24]. Consistent with the proteomics analysis, serum S100A9 levels were significantly decreased upon HBO therapy, as evident from the ELISA analysis.

### 2.2. Increased Serum S100A9 Level Was Associated with Inflammation, Vessel Injury and Bone Turnover

Vessel injury, bone damage, and excessive inflammation occur in the development of ONFH. The analysis of serum S100A9 levels prompted us to examine whether the protein has anything to do with the deleterious reactions. Thirty-eight patients with ONFH and 14 healthy controls were enrolled (Table 2). Of the patients, 12, 5, 11, and 10 patients were diagnosed with stage I, II, III, and IV ONFH, respectively, as radiographically diagnosed using the Ficat and Arlet system. Serum S100A9 levels (Figure 2A), along with angiogenic factor VEGF (Figure 2B), vessel injury marker vascular cell adhesion molecule-1 (VCAM-1) (Figure 2C), inflammatory cytokine interleukin-6 (IL-6) (Figure 2D), and bone resorption marker tartrate-resistant acid phosphatase 5b (TRAP5b) levels (Figure 2E) in sera were significantly increased in the ONFH group, whereas serum bone formation marker osteocalcin levels (Figure 2F) were significantly reduced in the ONFH group as compared to the healthy controls.

### 2.3. Increased S100A9 Levels Were Correlated with the Severity of ONFH

Next, we broke down the analysis of serum S100A9 to understand how much this serum protein may circulate in patients with different stages of ONFH. Of interest, serum S100A9 levels were significantly increased with the Ficat and Arlet stages. Patients with stage III and IV ONFH had the greatest increase in S100A9 levels (Figure 3A). Serum S100A9 levels were also significantly increased in patients with history of glucocorticoid medication (*n* = 16) and alcohol consumption (*n* = 13) (Figure 3B). Moreover, serum S100A9 was a powerful indicator for discriminating ONFH, as evident from the receiver operative characteristic (ROC) curve analysis, where the area under curve (AUC) was 0.9258 (*p* < 0.001) (Figure 3C).

### 2.4. Strong S100A9 Immunostaining and Hypovasculature Histopathology in ONFH

We conducted immunohistochemical analysis to characterize which compartment of osteonecrotic tissue S100A9 distributes. Femoral head specimens were harvested from patients with Ficat and Arlet stage IV ONFH and patients with displaced femoral head fractures who required total hip arthroplasty. Thrombosed vessels (Figure 4A), marrow adipose (Figure 4B), and fibrotic tissue (Figure 4C), along with osteocytes in cortical bone and inflammatory cells, showed strong S100A9 immunostaining as compared to the non-ONFH group (Figure 4D). Consistently, the number of S100A9-immunostained injured vessels, fat cells, osteocytes, fibroblasts, and inflammatory cells were significantly upregulated in the ONFH group (Figure 4E).

In addition, very few vessels developed in the ONFH group, as evident from the weak immunoreactivity for endothelial cell marker CD31 (Figure 5A) and capillary vessel marker vWF (Figure 5B), along with significant decreases in the CD31-immunostained (Figure 5C) and vWF-immunostained vessels (Figure 5D), which is indicative that S100A9 may be deleterious to vessel integrity in the development of ONFH.

### 2.5. S100A9 Inhibits Angiogenesis of Vessel Endothelial Cells and Aortic Rings

Given that increased S100A9 levels were correlated with a decreased vessel formation histopathology in ONFH, we wondered what role S100A9 may play in this event. To this end, human vessel endothelial cells were incubated in ONFH serum with or without the S100A9 antibody. For positive controls, cell cultures were incubated in a VEGF recombinant protein and showed intensive tube morphology (Figure 6A), along with significant increases in tube formation (Figure 6B) as compared to the vehicle group. The ONFH serum and S100A9 recombinant protein significantly reduced tube formation, whereas adding S100A9 significantly downregulated the ONFH serum-mediated loss of tube formation (Figure 6A,B). Likewise, VEGF-treated rat aortic rings displayed long and intensive micro-vessel outgrowth morphology (Figure 6C) together with significant increases in micro-vessel length (Figure 6D), whereas ONFH serum and S100A9 recombinant protein downregulated microvessel formation. The S100A9 antibody significantly mitigated the ONFH serum inhibition of micro-vessel outgrowth of aortic rings (Figure 6C,D).

## 3. Discussion

The severe destruction of femoral head microarchitecture, like subchondral bone breakdown, trabecular bone lacunae loss and marrow adiposity, etc., is a notable feature of ONFH, ultimately leading to the collapse of the femoral head [29,30]. The large number of bone cells in the lesion that progress toward apoptosis also accelerate the development of hip disease [31]. While ischemic injury is known to provoke ONFH, the molecular mechanistic underlying vessel integrity loss is poorly characterized. Collective analysis of this study reveals the relevance of S100A9 to the development of ONFH. S100A9 ramped up hypovasculature histopathology through repressing the angiogenesis of vessel endothelial cells. To our best knowledge, the intriguing investigation of this study is the first indication, shedding new light on the molecular pathologic events underlying ONFH. Robust evidence also highlights an emerging biochemical indication of S100A9 discriminating ONFH.

Many serum proteins in the ONFH group, like inflammatory cytokine IL-6, osteoclast marker TRAP5b, and osteoblast marker osteocalcin, were significantly higher than the healthy controls. The analysis of this study was in agreement with other groups′ studies, showing that upregulated IL-6 in serum [27], synovial fluid [32], and articular cartilage of osteonecrotic hip [33] and excessive bone remodeling [34] take place during human ONFH. While avascular destruction is a hallmark of ONFH, vessel remodeling in the hip disease remains uncertain. For example, angiogenic reactions, together with osteogenesis-regulatory microRNA, are downregulated in ONFH [35]. Osteonecrotic bone tissue shows impaired blood flow and vascular function [36]. On the contrary, articular cartilage in human ONFH displays increased VEGF immunostaining [37]. VEGF mRNA expression and vessel formation are augmented in experimental ONFH in swine [38,39]. Analysis in this study uncovered an upregulated serum VEGF level, together with hypovascularization histopathology as evident from weak CD31 and vWF immunostaining in ONFH. While the speculation is that angiogenic activity may depend on different compartments of osteonecortic lesions, an increase in the vessel injury marker VCAM-1 level, as shown in this study, further underpins the histopathologic analysis of decreased vessel formation. On the other hand, multiple biological reactions, like inflammation, vessel remodeling, and bone turnover, etc., simultaneously dysregulated in the femoral head microenvironment also explained the complex nature of ONFH. The investigations of vessel underdevelopment reasoned us to elucidate which molecule may modulate this reaction.

Profound investigations unraveled a significant increase in serum S100A9 level, which was correlated with the severity of ONFH. Of note, this serum protein appeared to be a sensitive indication for discriminating ONFH. Increasing evidence has shown the S100A9 exacerbation of degenerative joint diseases. Mice deficient in S100A9 show low mobilization of monocytes into synovium compartment in knees with collagenase-induced osteoarthritis [40]. S100A9 increases inflammatory reaction and cartilage breakdown in human knee joints [41]. This study showed immunohistochemical analysis that fatty marrow, fibrotic tissue and inflammatory cells, etc. in osteonecrotic tissue elicited strong S100A9 immunoreaction, which was consistent with the analysis of serum S100A9 overproduction. Moreover, S100A9 is known to escalate inflammatory cytokine IL-6 production in osteoarthritic joints [42] and monocytic cells [43]. Robust evidence of increased serum S100A9 and IL-6 in ONFH offers a new molecular insight into tissue deterioration occurring in ONFH. Given that S100A9 flared up deleterious activities, a significant reduction in serum S100A9 level in ONFH upon HBO therapy may explain the alleviating effect the therapy has elicited to this hip disorder.

Of interest, S100A9 is known to modulate the tube formation of the CD31+ and CD34+ endothelial progenitor cells [44]. It also increases the permeability and apoptotic program of endothelial cells [18] and artery dysfunction [45]. In vitro analysis has revealed the angiogenesis-inhibitory actions of S100A9 in the ONFH microenvironment as an antibody blockade of S100A9, improving the ONFH serum repression of tube formation of vessel endothelial cells and vessel outgrowth of aortic rings. We do not rule out the possibility that S100A9 may affect osteoclastic differentiation [46], resulting in a significant increase in the serum TRAP5b level in ONFH. The limitation of this study should be acknowledged, and further investigations are warranted to understand how S100A9 is increased and how the molecule deteriorates vessel integrity in the development of ONFH. This study conveys a new mechanistic underlying vessel damage, which provokes avascular necrosis during ONFH.

Taken together, dazzling evidence sheds light on the clinical relevance of S100A9 to the development of ONFH. S100A9 mediates hypovasculature in osteonecrotic tissue through hindering vessel formation. This study also highlights serum S100A9 as a sensitive indication discriminating ONFH.

## 4. Materials and Methods

### 4.1. Patients and Healthy Controls

All protocols for the harvest and detection of human specimen followed the ethical guidelines and were approved by the Institutional Review Board of Chang Gung Memorial Hospital (IRB #98-3926A3, 04 March 2010). Written informed consent was obtained from all patients and healthy participants. The exclusion criteria were a history or diagnosis of Paget′s disease, renal osteodystrophy, hyperparathyroidism, hypoparathyrodism, or bone metastasis. Thirty-eight patients with Ficat and Arlet stage I, II, III and IV ONFH were enrolled in the ONFH group, and 14 healthy controls were enrolled in the non-ONFH group (Table 2).

### 4.2. Two-Dimensional Gel Electrophoresis and Tandem Mass Spectrometry

The two-dimensional gel electrophoresis of cell lysates was performed to characterize protein expression profiles, as previously described [47]. In brief, 10 mL of peripheral blood in patients with Ficat and Arlet stage II ONFH, before and after HBO therapy, were harvested and processed to isolate the nucleated cells. Cell lysates of 10^7^ nucleated cells were extracted by mixing with 50 μL of PRO-PREP^®^ buffer (iNtRON Biotechnology Co., Burlington, MA, USA), desalted using PlusOne 2-D Clean-Up kits (Amersham, Picataway, NJ, USA), and centrifuged to collect pellets. The pellets were mixed with a rehydration buffer (2% CHAPS, 1% dithiothreitol, 0.5% IPG (pH 4–7), 8 M urea) (Amersham, Picataway, NJ, USA). Then, 150 μg of cell lysates were pipetted onto immobiline™ strips (pH 4–7, 13 cm) and isoelectric focused using the Ettan™ IPGphor II/3 system (GE Healthcare Bio-Sciences AB, Chicago, IL, USA) [44]. The isoelectric focused proteins in the strips were equilibrated in a solution containing 50 mM Tris–HCl (pH 8.8), 30% glycerol, 2% SDS, 0.25% iodacetamide, and 8 M urea, and were followed by SDS-PAGE analysis. The two-dimensional gels were silver stained (Amersham Pharmacia, Picataway, NJ, USA) according to the maker′s instructions. The silver-stained gels were scanned using the Amersham ImageScanner (Amersham BioScience Inc., Picataway, NJ, USA). The intensities of protein spots were compared using the Bio-Rad Proteoweaver 2-D Analysis Software Version 4.0, according to the manufacturer′s instructions. Protein spots with a difference in spot intensity greater than a 2-fold change were selected, as previously described [48].

### 4.3. Tandem Mass Spectrometry

The selected protein spots were excised, mixed with 20 ng/mL trypsin (Amersham, Picataway, NJ, USA), incubated at 37 °C for 16 h, and extracted with 1% trifluoroacetate. The peptide footprint of the extract was characterized using Ultraflex™ matrix-assisted laser desorption ionization time-of-flight tandem mass spectrometry (Bruker Daltonics, Hamburg, Germany). Match peptide mass were performed using MASCO bioinformatics. The criteria of candidate protein were that the peptide mass showed MASCOT score ≥ 65, along with a sequence coverage of ≥ 20% of the matched peptides of the Swiss-Port database. The MOWSE score ≥ 39 of the match peptide was considered as homology of the match protein [48].

### 4.4. ELISA Quantification of Serum Proteins

Serum S100A9 (CV, 7.2%; MyBiosources, San Diego, CA, USA), VEGF (CV, 5/2%; R&D Systems, Minneapolis, MN), IL-6 (CV, 4.2%; R&D Systems, Minneapolis, MN, USA), VCAM-1 (CV, 3.7%; R&D Systems, Minneapolis, MN, USA), osteocalcin (CV, 4.8%, novex^®^, Life Technologies, Carlsbad, CA, USA), and tartrate-resistant acid phosphatase-5b (CV, 5.1%; MyBiosources, San Diego, CA, USA) levels were measured using respective ELISA kits, according to the manufacturers’ manuals. The standard curves of various concentrations of authentic recombinant proteins of interest vs. absorbance detected by spectrophotometry were plotted.

### 4.5. Immunohistochemistry

Femoral heads from 10 patients with ONFH and 6 patients with displaced femoral neck fracture who required total hip arthroplasty were collected upon osteotomy. Skeletal specimens (1.5 cm × 1.5 cm × 1.5 cm) were dissected from the center of femoral head, as previously described [31]. The specimens were decalcified and embedded in paraffin for microdissection. S100A9, CD31, and vWF immunoreaction in sections were probed using S100A9, CD31 and vWF antibodies (Abcam, Cambridge, UK) and BioGenex immunohistochemistry detection kits (BioGenex, Fremont, CA, USA). Sections probed with IgG were designated as negative controls. Osteonecrosis signs, like bone lacuna loss, fatty marrow, fibrosis, thrombosis, and vessel injury, etc., were examined using a Zeiss microscope. Histological images were captured using the Zeiss Image Analysis unit (Zeiss, Oberkochen, Germany). Vessels, fibroblasts, fat cells, osteocytes and inflammatory cells showing S100A9 immunostaining, as well as vessels displaying CD31 and vWF immunoreaction in each field were counted. In total, 3 fields of each section and 2 sections of each specimen were randomly selected for histomorphometry.

### 4.6. Analysis of Angiogenesis of Human Vessel Endothelial Cells

The angiogenesis of human vessel endothelial cells incubated in ONFH serum was performed using BD BioCoat™ angiogenesis kits (BD Biosciences, San Jose, CA, USA), as previously described [49]. In brief, human umbilical vessel endothelial cells (5 × 10^4^ cells/well, 48 well-plates; American Type Culture Collection) were incubated in ONFH serum, with or without 1 μg/mL S100A9 antibody, IgG (Abcam, Cambridge, UK), 250 ng/mL S100A9 (R&D Systems, Minneapolis, MN, USA) and incubated in a humidified incubator at 37 °C for 6 h. In a subset of experiments, the cells were incubated in 250 ng/mL VEGF as positive controls. Tube morphology was evaluated using an inverted microscope (Zeiss, Oberkochen, Germany). The number of tubes in each field was counted. Six fields in each well, 3 wells in each specimen and 4 specimens were randomly selected for quantification.

### 4.7. Analysis of Vessel Formation of Rat Aortic Rings

Experimental animal use was approved by IACUC of Kaohsiung Chang Gung Memorial Hospital (Affidavit No. 2012122901). The protocols for vessel outgrowth of aortic rings were performed, as previously described [49]. In brief, 6 male Sprague-Dawley rats (3 months old) were euthanatized. The thoracic aortae were dissected and transversely cut into ring-like specimens under a surgery microscope in an aseptic condition. Aortic rings were put onto culture wells containing a mixture of 300 μL of Matrigel with 300 μL of MCDB131 medium (Life Technologies, Carlsbad, CA, USA) with 300 μL ONFH serum, with or without 1 μg/mL S100A9 antibody, IgG, 250 ng/mL S100A9 or 250 ng/mL VEGF, and incubated at 37 °C for 1 week. The length of vessel outgrown from the aortic rings was measured using a Zeiss microscope and the aforementioned Image Analysis System. In total, 9 fields in 3 aortic rings for each ONFH serum specimen and 6 ONFH serum specimens were selected for measurement.

### 4.8. Statistical Analysis

Analysis was expressed as mean ± standard error. The difference between groups were analyzed using an independent-sample t-test. Differences in gender were analyzed using a chi-square test. Data adjustment of age and gender were performed using a logistic regression test. Differences among groups were examined using an ANOVA test and followed by a Bonferroni post hoc test. The correlations among the serum protein were analyzed using Spearman′s correlation test. ROC curves and AUC values were calculated using SPSS software, according to the maker′s instruction. Analysis results with a *p*-value < 0.05 were considered statistically different.

## Figures and Tables

**Figure 1 ijms-20-05786-f001:**
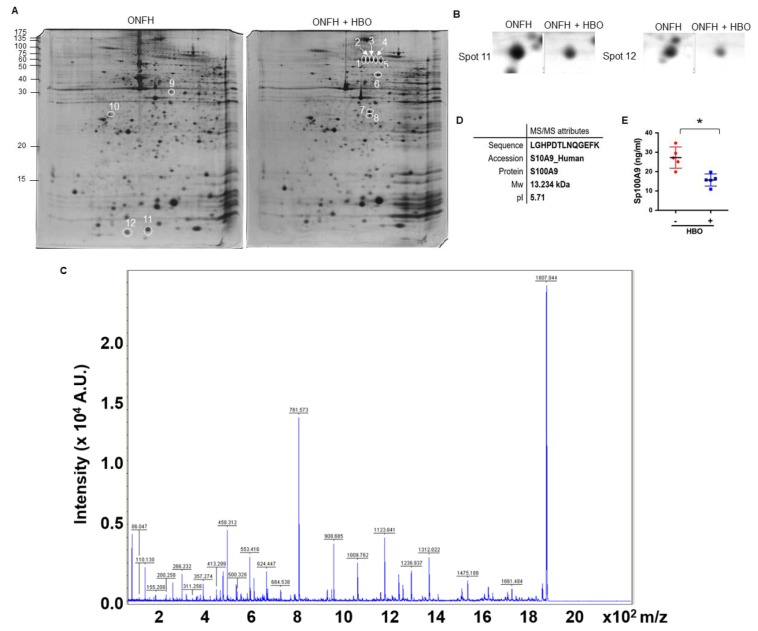
Analysis of S100 calcium binding protein A9 (S100A9) levels in peripheral blood cells and serum. Two-dimensional SDS-PAGE gel images of peripheral blood nucleated cells in patients with and without HBO therapy (**A**). Decreased S100A9 protein spots in the HBO group (**B**). Tandem mass spectrometrograms (**C**) and attributes (**D**) of S100A9. Decreased serum S100A9 levels in patients with osteonecrosis of the femoral head (ONFH) upon HBO therapy (**E**). Data are expressed as mean ± standard errors calculated from 5 patients. * *p* < 0.05. ONFH, osteonecrosis of the femoral head; HBO, hyperbaric oxygen. Red circles, patients with ONFH before HBO therapy; Blue squares, patients with ONFH upon HBO therapy.

**Figure 2 ijms-20-05786-f002:**
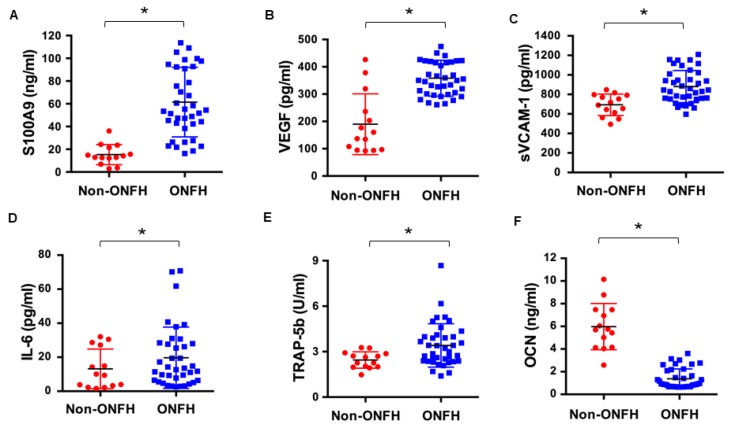
Serum protein levels in patients with ONFH and healthy controls. Significant increases in S100A9 (**A**), VEGF (**B**), sVCAM-1 (**C**), IL-6 (**D**) and TRAP-5b (**E**) levels along with significant reductions in serum osteocalcin levels (**F**) in the ONFH group. Data are expressed as mean ± standard errors. * *p* < 0.05. Data are expressed as mean ± standard errors calculated from 38 patients with ONFH and 14 healthy controls. * *p* < 0.05.

**Figure 3 ijms-20-05786-f003:**
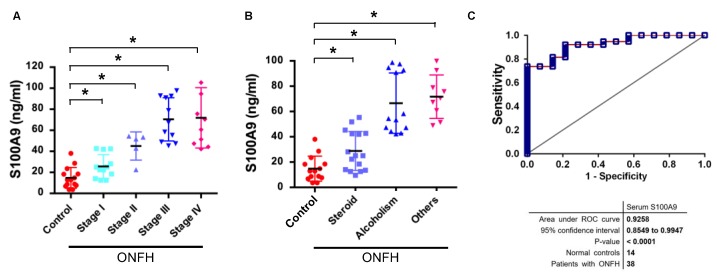
Correlation of serum S100A9, Ficat stages and etiological causes of ONFH. Serum S100A9 levels were increased with Ficat and Artlet stages of ONFH (**A**). Significant increases in serum S100A9 in patients with history of being glucocorticoid medication and alcohol consumption (**B**). Receiver operative characteristic (ROC) curve of serum S100A9 levels for discriminating ONFH (**C**). Data are expressed as mean ± standard errors calculated from 12, 5, 11, and 10 patients diagnosed with stage I, II, III, and IV ONFH and 14 healthy controls. * *p* < 0.05.

**Figure 4 ijms-20-05786-f004:**
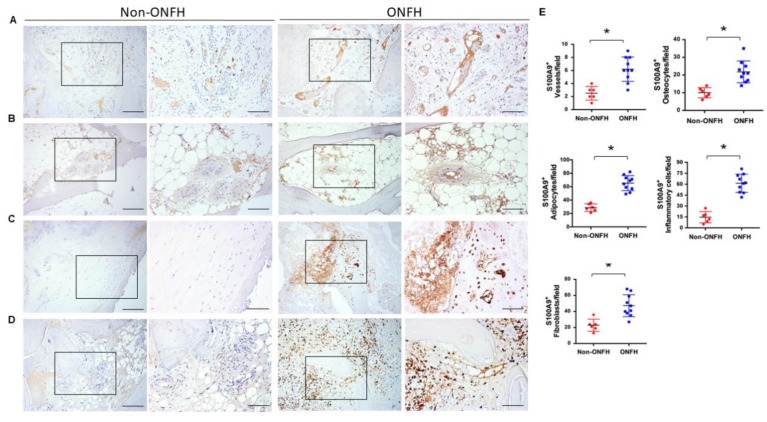
Immunohistochemical analysis of S100A9 in femoral head tissue. Injured vessels (**A**), marrow adipose (**B**), fibrotic tissue (**C**), and osteocytes in cortical bone and inflammatory cells (**D**), showed strong S100A9 immunostaining, along with significant increases in S100A9-immunostained vessels, fat cells, fibroblasts, osteocytes and inflammatory cells (**E**). Scale bares, 100 μm (panels 1 and 3) and 50 μm (panels 2 and 4). Data are expressed as mean ± standard errors calculated from 10 patients with ONFH and 6 patients with a femoral neck fracture who required total hip arthroplasty. * *p* < 0.05.

**Figure 5 ijms-20-05786-f005:**
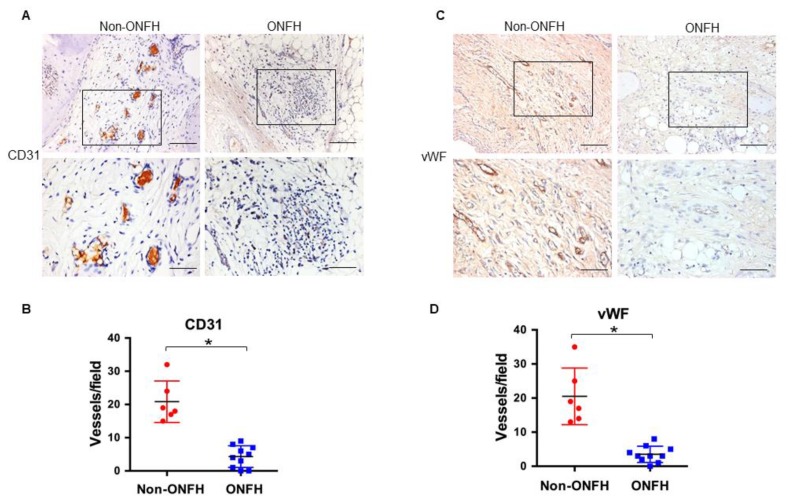
Immunohistochemical analysis of CD31 and vWF in femoral head. Weak CD31 (**A**) and vWF (**B**) immunostaining along with significant decreases in CD31-immunostained (**C**) and vWF-immunostained (**D**) vessels in the ONFH group. Scale bars, 100 μm (upper panels) and 50 μm (lower panels). Data are expressed as mean ± standard errors calculated from 10 patients with ONFH and 6 patients with displaced femoral neck fracture who required total hip arthroplasty. *, *p* < 0.05.

**Figure 6 ijms-20-05786-f006:**
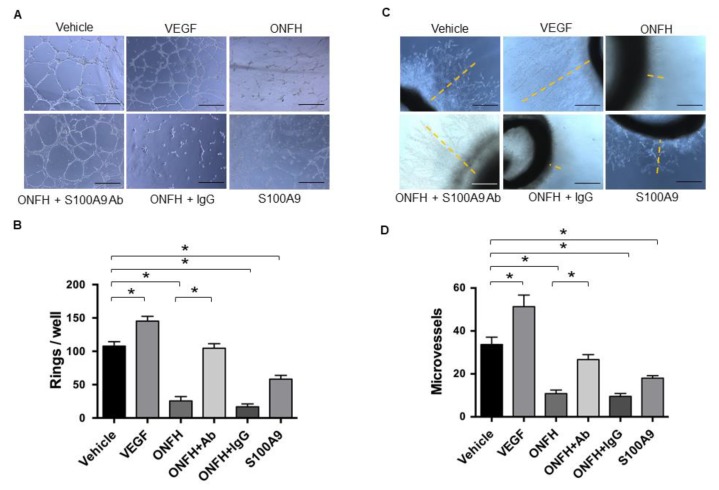
Analysis of tube formation of vessel endothelial cells and vessel outgrowth of aortic rings. Tube-like morphology of human vessel endothelial cells incubated with or without ONFH serum, S100A9 antibody and S100A9 (**A**). Scale bar, 50 μm. VEGF and antibody blockade of S100A9 attenuated ONFH serum increased tube formation, whereas S100A9 repressed angiogenesis of vessel endothelial cells (**B**). Vessel sprouting morphology of rat aortic rings (**C**). Scale bars, 25 μm. ONFH serum and S100A9 decreased vessel formation, whereas S100A9 antibody blockage improved vessel spouting (**D**). Data are expressed as mean ± standard errors calculated from 6 patients with ONFH.

**Table 1 ijms-20-05786-t001:** Tandem mass spectrometric analysis of serum protein.

Spot	Protein Name	Mw	pI	Accession
1	Filamin A (FLNA protein)	88534	5.93	gi|15779184
2	Filamin A (FLNA protein)	88534	5.93	gi|15779184
3	Filamin A (FLNA protein)	88534	5.93	gi|15779184
4	Gelsolin isoform B	80591	5.58	gi|38044288
5	Filamin A (FLNA protein)	88534	5.93	gi|15779184
6	Gelsolin isoform B	80591	5.58	gi|38044288
7	Protein disulfide isomerase-associated 3 isoform 1	54929	6.42	gi|114656687
8	Annexin III chain A	36309	5.63	gi|157830132
9	Capping protein (actin filament) muscle Z-line	29277	6.45	gi|55665440
10	Actin beta (ACTB protein)	40194	5.55	gi|15277503
11	S100A9	13234	5.71	S10A9_HUMAN
12	S100A9	13234	5.71	S10A9_HUMAN

**Table 2 ijms-20-05786-t002:** Demography of healthy volunteers and patients with ONFH.

	Healthy Controls	ONFH				
		Total	Stage I	Stage II	Stage III	Stage IV
Patients	14	38	12	5	11	10
Age	34.5 ± 5	50.4 ± 17.8	40.9 ± 10.9	58.2 ± 10.4	47.8 ± 12.2	57.8 ± 15
	(28–47)	(25–78)	(25–63)	(40–70)	(29–78)	(37–76)
Gender						
Males	7	18	2	3	7	6
Females	7	20	10	2	4	4

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
