# Peer review of "S100 Calcium Binding Protein A9 Represses Angiogenic Activity and Aggravates Osteonecrosis of the Femoral Head"

_ijms, 2019, doi:10.3390/ijms20225786_

Round 1
Reviewer 1 Report
The authors study the role of a calcium binding protein S100A9 and other serums in osteonecrosis of the femoral head (ONFH). The study is interesting and should be published after minor revision. In this study, proteomics analysis revealed a decrease in serum S100A9 level in patients with ONFH upon hyperbaric oxygen therapy. They also find out the regulation of other serums during ONFH. All together, the study states that increased S100A9 levels is relevant to the development of ONFH. S100A9 appears to provoke avascular damage, ultimately accelerating femoral head deterioration through reducing angiogenesis.
There are various typos in the manuscript. For example, third line of abstract. ".....understood is which role S100A9 my play in ONFH". Please give it a thorough read and check minutely.
Author Response
Thank you for bringing this to our awareness. We corrected the typos throughout the text.
Reviewer 2 Report
The authors have shown a decrease in serum S100A9 level in patients with ONFH upon hyperbaric oxygen therapy. They showed that thrombosed vessels, fibrotic tissue, osteocytes and inflammatory cells displayed strong S100A9 immunoreactivity in osteonecrotic lesion. In vitro, ONFH serum and S100A9 inhibited tube formation of vessel endothelial cells and vessel outgrowth of rat aortic rings, whereas antibody blockade of S100A9 improved angiogenic activities. They concluded that increased S100A9 levels is relevant to the development of ONFH.
There are however several issues that need to be addressed, before this manuscript is acceptable for IJMS:
Author Response
The association between S100A9 and ONFH is interesting. However, the mechanistic insight is not good enough that is critical. What is the receptor of S100A9? The vessel endothelial cells express this receptor? Deletion of the receptor could rescue the S100A9-mediated inhibition of tube formation? What is the signal between the receptor and tube formation? The signal is activated or suppressed by S100A9?
Authors’ response: Thank you, point taken. We re-wrote the sentences to interpreted the molecular events underlying S100A9 regulation of angiogenic activity of endothelial cells in the revised version (Lines 62-64), which now read as follows:
Toll-like receptor 4 (TLR4) mediates the S100A9 aggravation of inflammatory reaction and ischemia/reperfusion-mediated cardiovascular damage [16,17]. Inhibition of TLR4 signaling in vessel endothelial cells attenuates the S100A9-induced angiogenesis loss [18] through NF-κB, signal transducer and activator of transcription 3 (STAT3) and mitogen-activated protein kinase (MAPK) pathways [19].
If the ONFH serum suppresses tube formation in vitro, why are only bone vessel affected? Other vessels are also affected?
Authors’ response: We acknowledged the limitation of this study in the revised version (Lines 224-228), which now read as follows:
We do not rule out the possibility that S100A9 may affect osteoclastic differentiation [46], resulting in a significant increase in serum TRAP5b level in ONFH. The limitation of this study should be acknowledged is further investigations are warranted to understand how S100A9 is increased and how the molecule deteriorates vessel integrity in the development of ONFH.